# Senior Theater Projects: Enhancing Physical Health and Reducing Depression in Older Adults

**DOI:** 10.3390/ijerph21101289

**Published:** 2024-09-26

**Authors:** Ayuto Kodama, Nobuko Watanabe, Hitomi Ozawa, Shinsuke Imamura, Yoko Umetsu, Manabu Sato, Hidetaka Ota

**Affiliations:** 1Advanced Research Center for Geriatric and Gerontology, Akita University, Akita 010-8543, Japan; ay-kodama@med.akita-u.ac.jp; 2Department of Occupational Therapy, Graduate School of Medicine, Akita University, Akita 010-8543, Japan; 3General Incorporated Association Warabi-za, Warabi 014-1113, Japan; toyo@warabi.or.jp (N.W.); oulan1915@gmail.com (H.O.); shinsuke@warabi.or.jp (S.I.); 4Integrated Community Support Center, Public Health and Welfare Department, City Hall of Yokote, Yokote 013-8601, Japan; umetsu-yoko-a@city.yokote.lg.jp (Y.U.); sato-manabu-a@city.yokote.lg.jp (M.S.)

**Keywords:** theater intervention, depression symptoms, physical frailty

## Abstract

Background: The aim of this study was to clarify the effects of a theater training program intervention on the physical and cognitive functions in community-dwelling older people. Methods: Of the 59 participants, 30 were the control group, and 29 were the intervention group. We assessed physical and mental/cognitive functions and criteria of physical frailty. Results: Statistical analysis showed that the usual walking speed (UWS) (*p* < 0.01), grip strength (GS) (*p* < 0.01), and GDS-15 (*p* < 0.05) improved significantly in the intervention group, whereas the Geriatric Depression Scale short-form (GDS-15) (*p* < 0.01) worsened significantly in the control group. Cognitive function was not significantly different between the two groups. Physical frailty was unchanged in the control group but significantly improved in the intervention group (*p* < 0.05), and a significant interaction was found for GDS-15 in ANOVA (F = 5.76, *p* < 0.05). Conclusions: The results of this study suggest that a theater intervention for the older adults may be effective in preventing and improving depression and physical frailty in old age.

## 1. Introduction

Frailty in older adults is an increasingly recognized clinical syndrome marked by decreased reserve and diminished resistance to stressors due to declines in physical, cognitive, and emotional function, which leads to increased risks of adverse health outcomes [1]. This condition is commonly characterized by symptoms such as sarcopenia, exhaustion, low physical activity, slowed walking speed, and unintentional weight loss. These criteria, known as Fried’s criteria, highlight the multifaceted nature of frailty and underscore the urgency for targeted intervention strategies. Some study results showed that frailty is an independent indicator of poor prognosis in older adults, particularly with increased risk of falls, hospitalization, and death. It has been concluded that the assessment of frailty is important in the development of intervention strategies and that early detection and treatment may contribute to the maintenance of health in older adults [2]. The European Union has placed specific importance on defining frailty, as frail persons are high users of community resources, hospitalization, and nursing homes. It is assumed that early intervention with frail persons will improve quality of life and reduce costs of care [3,4].

Current research supports the effectiveness of various interventions for managing physical frailty in older adults. In particular, resistance training emerges as the most beneficial method, improving muscle strength and functional status and outperforming other interventions such as usual care and pharmacotherapy [5]. Binder et al. reported that balance and strength training exercises performed on 117 older adult subjects resulted in significant improvements in muscle strength, balance, and quality of life in the exercise group compared to the control group [6]. Moreover, the integration of nutritional therapy, particularly high-protein diets and vitamin supplements, along with physical exercise, has shown promise in addressing multiple facets of frailty [7,8]. Despite these advancements, the evidence on the benefits of combining different domain interventions remains sparse and inconclusive, necessitating further exploration into their collective impact on frailty.

On the other hand, Noice et al. investigated changes in cognitive function after a short-term theatrical intervention in older adults [9,10,11]. These studies compared participants in a group that received theater training (*n* = 44), visual arts training (*n* = 44), or no treatment (*n* = 36) in nine 90 min sessions over a one-month period. The results showed that the performance of the theater intervention group showed significant improvement in word memory and problem-solving tasks. Other previous studies suggest that theater-based interventions may improve cognitive functions and problem-solving abilities, which are crucial for maintaining independence in daily activities among older adults [12,13]. Stern et al. have also shown that artistic practices promote cognitive constructs [14], and artistic practices such as theater have been reported to promote health in a variety of older adults [15]. Moreover, while it has been reported that engagement in arts activities may help to reduce sedentary behavior, a well-established predictor of cardiovascular health and immune function, few reports have been found on the use of theater intervention to improve physical frailty [16,17]. However, theater interventions have shown that combining physical movements with cognitive stimulation contributes to the prevention of dementia and the maintenance or improvement of physical function [18]. Additionally, theatrical and musical activities have been reported to improve the mental health of elderly individuals by providing emotional outlets and stabilizing psychological well-being [19].

This study aims to explore an innovative approach to mitigating physical frailty through a senior theater project facilitated by the Warabi-za Association (Warabi-za). The Warabi-za, based in Akita Prefecture, is a renowned theater company founded in 1951. Known for integrating Japanese traditional culture into its performances, Warabi-za focuses on regional folklore, history, and customs, especially from the Tohoku region. Their productions highlight and preserve local culture while engaging modern audiences through musical theater. This research program was developed and supervised by Warabi-za. We have previously reported on a preliminary study of the effects of a theatrical intervention with older adults [20]. In that study, it was suggested that theatrical interventions are as effective as the multicomponent exercise that has been recommended in frailty prevention and that theatrical interventions are effective in strengthening the community of participants and improving social frailty. Unlike traditional interventions, this approach incorporates theatrical training, focusing on enhancing cognitive and physical functions through creative engagement rather than direct physical training. This research will assess the effectiveness of these theatrical interventions in a controlled environment and evaluate their potential as a comprehensive therapy for frailty.

## 2. Materials and Methods

### 2.1. Participants

This quasi-experimental study was conducted from October 2023 to March 2024. Potential participants aged 65 years and older were recruited through promotional materials distributed to the public in various regions of Akita Prefecture to obtain informed consent for their participation in the study. Participants were required to be able to walk independently and live at home without personal assistance. Exclusion criteria included “dementia”, “major depression”, “severe hearing or visual impairment”, “stroke”, “Parkinson’s disease”, “other neurological disorders”, “intellectual disability”, “certification for long-term care support due to disability under the Japanese public long-term care insurance system”, and “inability to complete cognitive tests at the baseline assessment”. The sample size was calculated using the G*Power 3.1.9.4 statistical power analysis program to estimate the number of participants required [21,22]. “Using G*Power to perform a sample size calculation for independent-samples *t*-tests (comparing means between control and intervention groups), with an allocation ratio of 4:1 favoring the control group, it was determined that a sample size of 160 participants in the control group and 40 participants in the intervention group would be required to detect a clinically meaningful difference between the groups. This calculation is based on a two-tailed test with a significance level (α) of 0.05, a statistical power of 80%, and a medium effect size (Cohen’s d) of 0.50”. Participants who expressed willingness to participate in the program and met the selection criteria were selected.

### 2.2. Intervention

Of the 75 participants who gave informed consent, they were assigned to either the control group or the theater intervention group. The control group was assessed twice, before and after the study period, but no interventions were performed during this time. The intervention group received 120 min of instruction from longtime theater professionals and teaching staff every two weeks for 6 months. The intervention group received 12 interventions as a total. Theatrical interventions led by theater professionals have been shown in numerous studies to effectively improve both the physical and mental functions of elderly individuals. Engaging in theatrical activities involves movements and vocal expressions that require full-body use, making it a form of exercise that enhances physical abilities. Specifically, improvements in balance, muscle strength, and flexibility can be expected, which help reduce the risk of falls and improve daily functional mobility.

Moreover, theatrical activities stimulate cognitive functions as they involve memorizing lines, choreographed movements, and emotional expressions. This process helps enhance memory and concentration, preventing cognitive decline. The need for spontaneous dialogue and decision-making during rehearsals and performances also sharpens attention and flexibility in thinking.

On the psychological side, theatrical interventions offer participants opportunities for emotional expression and social interaction, contributing to emotional relief and stress reduction. Taking on roles in performances boosts self-esteem and self-worth, enhancing life satisfaction and giving participants a greater sense of purpose. The theatrical intervention in one session consisted of three stages (Appendix A). The first stage focused on full-body stretching exercises and vocal exercises to warm up (Figure 1A). The second phase involved training to express emotions through body movements and facial expressions to release emotions (Figure 1B). In the third stage, the participants worked on their lines and practiced their dances during the first half of the period. They also brought their own props for use in the play or created their own if necessary. Each participant worked hard to find a role in which they excelled and collaborated to complete the play. In the second half, the participants practiced their plays, ensuring they had learned their lines and movements well and were in position with their counterparts (Figure 1C).

### 2.3. Outcome

We examined age, gender, education, and medication; moreover, dependence in daily activities and social activities was assessed using the Kihon checklist (KCL) [23] in the baseline. The KCL included Q1-25 items, with each score of the KCL indicating difficulty with the activity in the question, and a higher score of the checklist means a higher risk of requiring support for each domain. The outcomes measured, including cognitive functions, physical functions, and physical frailty, will be evaluated before the intervention (pre-test) and after 6 months (post-test). The physical performance assessments consisted of GS, UWS, and GDS-15 [24]. The GDS-15 presents a dichotomous response format (Yes/No) that makes them easy to administer, where conventionally in the GDS-15, a number equal to or greater than five positive items assumes mild depression and a number equal to or greater than 10 positive items assumes moderate to severe depression, and, hence, follow-up through clinical evaluation is recommended to confirm the diagnosis. Four cognitive subtests of the National Center for Geriatrics and Gerontology Functional Assessment Tool (NCGG-FAT) were also assessed for each participant. NCGG-FAT is a tool developed to assess cognitive functional health in older adults. It has been validated for reliability and validity in various studies. It correlates well with other established functional assessment tools and accurately reflects older adults’s cognitive functional status, supporting its use in both clinical and research settings [25]. Cognitive subtests are indicated as follows:

#### 2.3.1. Components of NCGG-FAT

The computerized multidimensional neurocognitive test was performed on an iPad (Apple, Cupertino, CA, USA) with a 9.7-inch touch display. The task instructions were presented with a letter size of at least 1.0 × 1.0 cm^2^ on the display. For this study, a trained operator supported each participant by setting up the tablet PC and running each test. Participants completed the NCGG-FAT subtests as follows:

#### 2.3.2. Tablet Version of Word Recognition (WR)

This test comprises two computerized tasks of immediate recognition and delayed recall. In the first task of immediate recognition, participants were instructed to memorize 10 words, each of which was displayed for 2 s on the tablet PC. After that, a total of 30 words, including 10 target and 20 distracter words, were shown to participants, and they were required to select the 10 target words immediately. This task was repeated for three trials. The average number of correct answers was recorded as a score ranging from 0 to 10. In another task, participants were asked to correctly recall the 10 target words after 20 min. The number of correctly recalled target words was scored ranging from 0 to 10. Finally, we calculated the sum score of the two tasks of immediate recognition and delayed recall.

#### 2.3.3. Tablet Version of Trail Making Test Version A (TMT-A) and Version B (TMT-B)

In the Trail Making Test Version A (TMT-A) task, participants were instructed to touch the target numbers in a sequence as rapidly as possible. Target numbers from 1 to 15 were randomly displayed on the tablet panel. In addition, the Trail Making Test Version B (TMT-B) instructions required participants to touch target numbers (e.g., 1–15) and letters in turn.

#### 2.3.4. Tablet Version of Symbol Digit Substitution Task (SDST)

In the Symbol Digit Substitution Task (SDST), nine pairs of numbers and symbols were shown in the upper part of the tablet display. A target symbol was shown in the center of the tablet panel, and selectable numbers were displayed at the bottom. Participants were asked to touch the number corresponding to the target symbol shown in the central part of the tablet display as rapidly as possible. The number of correct numbers within 90 s was recorded.

#### 2.3.5. Criteria of Physical Frailty

Frailty status was defined based on five dimensions of the Fried frailty index, including (i) shrinking, (ii) exhaustion, (iii) low level of activity, (iv) weakness, and (v) slowness, 0 for robust, 1–2 for prefrail, and 3–5 for frail, and the frailty [26].

### 2.4. Statistical Analysis

The participants who lacked data either at the baseline or after the intervention were excluded for analysis. The paired *t*-test was applied to compare the results of UWS, GS, WM, TMT-A & B, and SDST between the pre-test and post-test of this program for participants. The chi-squared test was applied for gender and physical frailty status.

For variables with significant differences yielded in the former comparison tests, the analysis of variance for split-plot factorial design was subsequently applied to examine interactions between the groups (control group and intervention group) and time (pretest/post-test) for main effects. The F-value and the effect size (η^2^) were calculated as statistics for the analysis of variance for split-plot factorial design. SPSS Version 27.0 for Windows (SPSS INC., Chicago, IL, USA) was used for the analysis, and the level of significance was set at *p* = 0.05.

## 3. Results

As shown in Figure 2, the final samples used for analysis consisted of 30 participants in the control group and 29 in the intervention group. Participant characteristics were compared to emphasize homogeneity between groups at baseline. The results showed a trend toward lower age and higher independence in the intervention group, but none of the differences were significant (Table 1). The paired *t*-test was used to analyze the differences between pre-test and post-test and revealed for the control group a significant decrease in the GDS-15 (*p* < 0.01) (Table 2). On the other hand, the intervention group showed a significant improvement in the GS (*p* < 0.05), UWS (*p* < 0.001), and GDS-15 (*p* < 0.05). In addition, the physical frailty status did not have a significant difference for the control group but had a significant improvement for the intervention groups (*p* < 0.05) (Table 3). Next, the UWS and GDS-15 had an interactive effect between the group and the time factors (Group Time interactions) (Figure 3 and Figure 4).

The control group showed a significant decrease in GDS-15 (*p* < 0.01). The intervention group showed a significant improvement in GS (*p* < 0.05), UWS (*p* < 0.001), and GDS-15 (*p* < 0.05).

The physical frailty status had a significant improvement for the intervention group (*p* < 0.05).

The UWS had an interactive effect between the group and the time factors (Group Time interactions) (** *p* < 0.01).

The GDS-15 had an interactive effect between the group and the time factors (Group Time interactions) (** *p* < 0.01).

## 4. Discussion

Our investigation revealed that participation in the theatrical training program led to significant improvements in UWS and GS among older adult participants. These improvements are similar to those observed in previous studies, where dynamic and rhythmic physical activities, such as those incorporated into multimodal exercise programs, have been shown to effectively increase lower extremity muscle strength and walking speed [5]. This alignment with existing research reinforces the validity of our findings and suggests that expressive, full-body movements in a theatrical context are comparably beneficial. In contrast, other research highlights the physiological mechanisms by which strength training specifically, not necessarily performance arts like theater, affects muscle function in older adults. This includes increased activation of prime mover muscles, enhanced co-activation of cooperative muscles, and reduced co-activation of antagonist muscles [27]. These findings suggest that traditional strength training might be more focused on these aspects than theatrical activities, which could offer a differing perspective on muscle function improvement. Despite these differences, our findings underscore the unique contribution of engaging in theater to physical health in older adults. The incorporation of expressive physical movements within the context of a meaningful narrative and emotional expression appears not only to provide physical benefits similar to those obtained through structured exercise but also introduces a novel and engaging approach to preventive intervention. This reaffirms the value of theatrical engagement as an effective intervention to enhance physical capabilities such as walking speed and muscle strength in older adults. In addition, a significant physical frailty improvement was observed after the theater-based intervention [28]. Previous theater-based interventions for older adults have focused on cognitive function with little effect on the physical aspect. On the other hand, this study suggests that the social and emotional connections gained through theater activities are a factor in preventing social isolation and supporting mental stability among the elderly. In this regard, it was confirmed that theater training not only improves physical function but also contributes to participants’ sense of spiritual fulfillment and strengthens their social networks. In this study, a new intervention effect using musicals was found.

Next, in our program, the process of participating in theater required not only physical activity but also active involvement of older participants in social and creative roles. Participants roles that included collaborative discussions and decisions about roles, props, and costumes promoted social engagement and created a cooperative community environment. We believe that such social interaction is important as it counteracts the isolation often experienced in old age and supports mental and emotional well-being. The literature in the field of gerontology supports the studies that maintaining social networks and engaging in meaningful social activities are crucial for healthy aging [29]. It has been observed that enhancing social engagement can buffer against the stress associated with physical and life changes in older adults, thus promoting a healthier mental state [30]. This aligns with our findings, where the act of creating and participating in a play seemed to instill a sense of purpose and community among participants, enhancing their mental stability. On the other hand, prior research has shown that moderate physical activity, if sustained long enough, can slow the progression of age-related frailty and depression [31,32,33,34]. This perspective emphasizes the importance of sustained physical activity in preventing age-related frailty and depression. However, this study suggested that the social and creative aspects of theater participation can also play an important role in supporting the mental health of older adults. Revisiting our findings, the collective and creative engagement in theater clearly benefited our participants by enhancing their mental stability and social connectivity, confirming the vital role of social interaction and emotional support in the process of aging.

Next, executive function and processing skills showed a trend toward improvement after the intervention. Theater training requires participants to memorize multiple lines and accompanying actions for each scene. It also requires an element of multitasking, as participants are sometimes asked to respond appropriately when other participants make mistakes in lines or actions. Noice et al. reported that theater-based interventions lead participants to experience the emotional states and heightened arousal associated with performing these tasks in front of peers while simultaneously keeping lines and actions in mind, so repeating such experiences may develop executive function and processing skills [13]. However, some research reported that more traditional cognitive exercises or direct cognitive training may offer more targeted benefits to cognitive health in older adult populations than complex social interactions like those found in theater [35,36]. These studies suggest that structured cognitive training might lead to more measurable improvements in specific areas of cognitive function. In the future, we would like to increase the sample size and compare the results with the effects of conventional training.

Finally, the manuscript highlights several important public health benefits of the theater intervention for older adults, especially in preventing frailty and improving both physical and mental health outcomes. Theater activities foster social interaction and emotional expression, which can mitigate the effects of social isolation, a common issue among the elderly. Social isolation is linked to various negative health outcomes, including increased mortality risk, and the study suggests that the intervention strengthens social bonds, contributing to both emotional well-being and overall life satisfaction. The manuscript provides strong evidence that integrating artistic and social activities into community-based health programs for the elderly could play a significant role in public health strategies aimed at enhancing both physical and mental well-being.

We need to consider the limitations of our research before we design future research. First, this study used a quasi-experimental design, which lacks the strict control that a randomized controlled trial (RCT) provides. The assignment of participants was not randomized, which could have introduced differences in baseline characteristics such as age and social or physical factors between the intervention and control groups. As a result, the ability to generalize the findings and establish a clear cause-and-effect relationship may be limited. Second, the sample size of this study was relatively small, with 30 participants in the control group and 29 in the intervention group, which limited the statistical power of the study. For example, although there were trends in the improvement of cognitive function, these differences were not statistically significant. A larger sample size may have allowed for more robust conclusions, particularly in detecting smaller or more nuanced effects. Third, another limitation is the absence of follow-up to assess the long-term effects of the theater intervention. Previous research on exercise interventions has shown that the benefits often diminish over time once the intervention ceases. Therefore, it is unclear whether the improvements observed in this study would persist beyond the immediate post-intervention period. Fourth, the theater intervention is multifaceted, involving not only physical movement but also emotional expression and social engagement. This makes it difficult to isolate which specific elements of the intervention contributed to the observed improvements. Additionally, other lifestyle factors such as diet, sleep, and routine activities could have influenced the results, but these were not fully controlled for in the study. Therefore, to determine the lasting impact of theater interventions, long-term follow-up assessments are necessary. Future studies should include follow-up periods of six months, one year, or even longer after the intervention to evaluate the durability of the observed improvements. This would provide valuable insights into whether the benefits of the intervention persist or diminish over time, as seen in other types of interventions. Moreover, future research should aim to disaggregate the different elements of theater interventions, such as physical movement, emotional expression, and social interaction, to better understand which components contribute most to the observed effects. This could be achieved by designing studies with multiple intervention groups, each focusing on a different aspect of the theater activity. Additionally, collecting data on participants’ lifestyle factors, such as diet, sleep, and physical activity, would allow for better control of potential confounders in the analysis.

## 5. Conclusions

This study provides compelling evidence that theatrical training programs offer significant benefits for older adults, both physically and mentally. Our findings demonstrate that engaging older adults in theater-based activities leads to notable improvements in walking speed and grip strength. Moreover, the participants not only experienced improvements in physical health but also showed enhanced social engagement and mental well-being. This dual benefit underscores the importance of addressing both the physical and psychological needs of the older adults. Our research also suggests that while theater can be a valuable component of cognitive care, it may need to be combined with more targeted cognitive exercises for maximum efficacy. By fostering an environment where physical activity, social interaction, and creative expression converge, theater-based programs can significantly enhance the quality of life for older adults, making a strong case for their inclusion in public health strategies aimed at this demographic.

## Figures and Tables

**Figure 1 ijerph-21-01289-f001:**
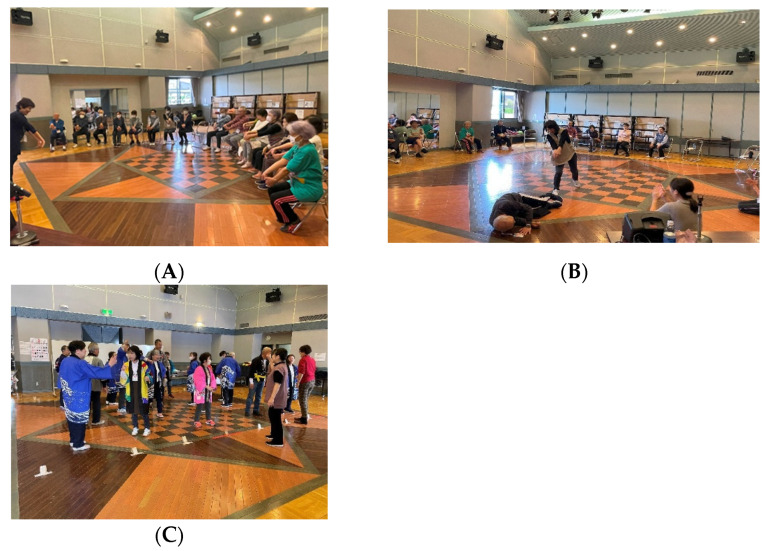
The theatrical intervention. (**A**) Mind and Body Stretching Exercises. (**B**) Emotional Release Training. (**C**) Theater Practice.

**Figure 2 ijerph-21-01289-f002:**
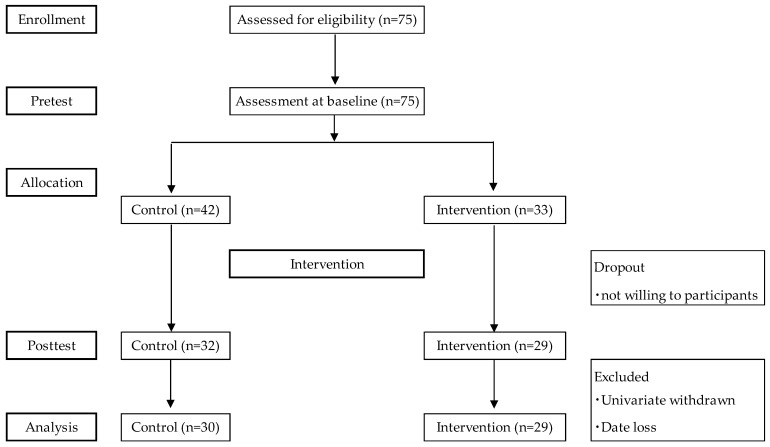
Flow diagram of screening.

**Figure 3 ijerph-21-01289-f003:**
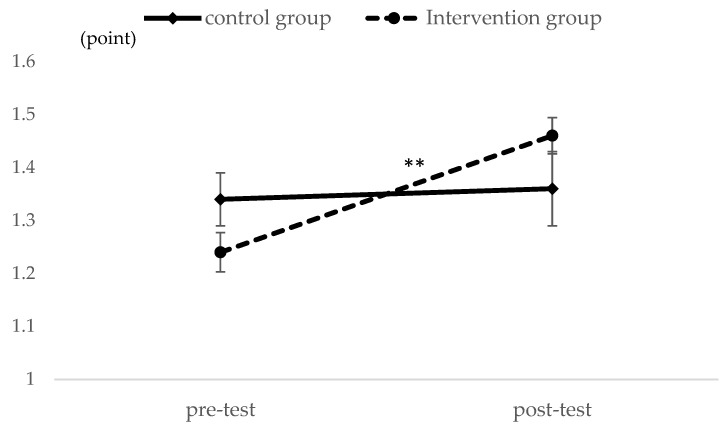
Main effect analysis of interaction regarding UWS between the control group and the intervention group. ** *p* < 0.01.

**Figure 4 ijerph-21-01289-f004:**
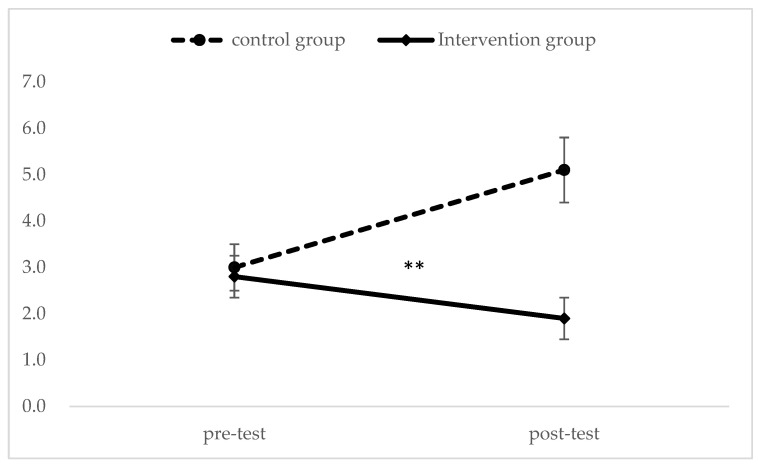
Main effect analysis of interaction regarding GDS-15 between the control group and the intervention group. ** *p* < 0.01.

**Table 1 ijerph-21-01289-t001:** Baseline characteristics and between group difference.

	Control Group		Intervention Group		*p* Value	Effect Size
	*n* = 30		*n* = 29			
	Mean	SD	Mean	SD		
Age (years)	75.6	5.5	73.1	5.8	0.061	−0.498
Gender (%female)	63.30		72.40		0.456	0.611
Height (cm)	156.3	6.8	155.6	6.5	0.664	−0.114
Weight (kg)	56.9	9.2	56.3	9.5	0.864	−0.045
Medication (*n*)	2.7	2.4	2.9	3.3	0.198	0.068
Education (years)	12.7	2.3	13.2	2.4	0.439	0.186
KCL (point)	4.5	3.0	3.6	3.4	0.647	0.120

The paired *t*-test (Age, Height, Weight, Medication, Education, and KCL), the χ^2^ test (gender). Abbreviations: SD, standard deviation; KCL, Kihon checklist. The basic characteristics of participants in the statistical analysis showed that there was no significant difference between the groups.

**Table 2 ijerph-21-01289-t002:** Differences in dependent variables between groups over time and interactions or main effects between the group.

	Control Group (*n* = 30)					Intervention Group (*n* = 29)													
	Pre-Test		Post-Test			Pre-Test		Post-Test			Interaction (Group × Time)			Main Effect (Group)			Main Effect (Time)		
	Mean	SD	Mean	SD	*p* Value	Mean	SD	Mean	SD	*p* Value	F (5.76)	*p* Value	η^2^	F (5.76)	*p* Value	η^2^	F (5.76)	*p* Value	η^2^
GS (kg)	26.0	7.9	26.3	7.3	0.526	25.2	1.3	26.6	1.4	0.009 **	0.778	0.382	0.013	0.047	0.830	0.001	9.206	0.004	0.139
UWS (m/s)	1.34	0.05	1.36	0.07	0.532	1.24	0.37	1.46	0.34	0.000 **	9.024	0.004 **	0.137	0.048	0.828	0.001	20.723	0.000**	0.267
WM (point)	13.6	0.6	12.8	0.8	0.153	14.4	0.5	15.0	0.5	0.282	2.735	0.104	0.046	2.870	0.096	0.048	0.016	0.901	0.000
TMT-A (s)	20.9	1.3	20.6	0.9	0.725	18.8	0.9	18.8	0.9	0.966	0.060	0.808	0.001	1.664	0.202	0.028	0.060	0.808	0.001
TMT-B (s)	40.8	4.4	41.9	4.0	0.738	32.1	2.4	28.6	1.8	0.099	0.412	0.524	0.027	6.003	0.017 *	0.095	0.412	0.524	0.007
SDST (point)	46.5	1.9	46.3	2.2	0.757	52.9	2.0	54.8	2.6	0.163	1.599	0.211	0.027	5.669	0.021 *	0.090	1.491	0.227	0.025
GDS-15 (point)	3.0	0.5	5.1	0.7	0.002 **	2.8	2.9	1.9	0.5	0.012 *	17.565	0.000 **	0.236	5.034	0.029 *	0.081	2.704	0.106	0.045

* *p* < 0.05, ** *p* < 0.01 the paired *t*-test, the analysis of variance for split-plot factorial design. Abbreviations: GS, Grip Strength; UWS, Usual Walking Speed; WM, Word Recognition; TMT-A, Trail Making Test-Version A and B; SDST, Symbol Digit Substitution Task; GDS-15, Geriatric Depression Scale short-form.

**Table 3 ijerph-21-01289-t003:** Comparison of physical frailty for the control group and the intervention group at the pre-test and the post-test.

	Control Group		*p* Value	Effect Size	Intervention Group		*p* Value	Effect Size
	Pre-Test	Post-Test			Pre-Test	Post-Test		
physical frailty (%)			0.723	0.323			0.018 *	0.417
robust	56.7	46.7			48.3	82.8		
pre-frailty	36.7	46.7			51.7	17.2		
frailty	6.7	6.7			0.0	0.0		

* *p* < 0.05, the χ^2^ test.

## Data Availability

Data are available on request due to restrictions, e.g., privacy or ethical. The data presented in this study are available on request from the corresponding author.

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
