# Peer review of "Senior Theater Projects: Enhancing Physical Health and Reducing Depression in Older Adults"

_ijerph, 2024, doi:10.3390/ijerph21101289_

Round 1
Reviewer 1 Report
Comments and Suggestions for Authors
Dear the authors,
Abstract
Rephrase to "Senior Theater Projects: Enhancing Physical Health and Reducing Depression in the Elderly" for a more streamlined title.
Define all abbreviations (e.g., UWS, GDS-15) when first mentioned.
Use consistent terminology, such as "the elderly" or "older adults," throughout the manuscript.
Include more specifics in the methods section about the nature of the assessments and the duration of the intervention.
Introduction
Focus on key points to make the introduction more concise.
Warabi-Za Association, Provide more details about the Warabi-Za Association and their role in the project.
Methods
Clearly state the inclusion and exclusion criteria in bullet points for easier reading.
Provide more details on how the sample size calculation was conducted using G*Power.
Include a brief rationale for why the theater intervention was chosen and how it was expected to impact the participants.
Adjust Scheme 1 and reorganize Figure 1 to present them as a single, cohesive figure.
Results
Add a brief description of the participants' demographics and any baseline comparisons to highlight the homogeneity between groups.
Ensure all p-values are reported with exact values, and include effect sizes where applicable to provide more insight into the magnitude of the findings.
Include more details in the table captions and figure legends to enhance understanding without having to refer back to the text.
Discussion and Conclusion
Provide more detailed comparisons with similar studies to strengthen the context of our results.
Offer a more detailed explanation of the limitations and how they might be addressed in future research.
Suggest more specific areas for future research, such as the impact of different types of theater-based interventions or long-term follow-up studies.
I believe these suggestions will significantly enhance the manuscript's clarity and impact.
Sincerely,
Comments on the Quality of English LanguageModerate editing of English language required.
Author Response
Response to Reviewer 1 Comments
Point 1: Abstract
Rephrase to "Senior Theater Projects: Enhancing Physical Health and Reducing Depression in the Elderly" for a more streamlined title.
Response 1: Thank you for supportive comments. As the reviewer suggested, we have changed the title.
Point 2: Define all abbreviations (e.g., UWS, GDS-15) when first mentioned.
Response 2: Thank you for supportive comments. As the reviewer suggested, we have modified all abbreviations. Page 1, Line 22
Point 3: Use consistent terminology, such as "the elderly" or "older adults," throughout the manuscript.
Response 3: Thank you for supportive comments. We modified consistent terminology, "older adults," throughout the manuscript.
Point 4: Include more specifics in the methods section about the nature of the
Response 4: Thank you for supportive comments. We have added description of the evaluation and intervention period.
(The intervention group received 120 minutes of instruction from longtime theater professionals and teaching staff every two weeks for 6 months. The intervention group received 12 interventions as a total. )
(KCL including Q1-25 items, each score of the KCL indicates difficulty with the activity in the question, and a higher score of the checklist means higher risk of requiring support for each domain. Page , Line )
(GDS-15 present a dichotomous response format (Yes/No) that makes them easy to administer, where conventionally in the GDS-15, a number equal to or greater than five positive items assumes mild depression and a number equal to or greater than 10 positive items assumes moderate to severe depression, and hence follow-up through clinical evaluation is recommended to confirm the diagnosis.)
Point 5: Introduction
Focus on key points to make the introduction more concise.
Response 5: Thank you for supportive reviews comments . As suggested by the reviewer, we have changed the Introduction.
Point 6: Warabi-Za Association, Provide more details about the Warabi-Za Association and their role in the project.
Response 6: Thank you for supportive reviews comments. As we have added description of the Warabi-Za.
(The Warabi-za based in Akita Prefecture, is a renowned theater company founded in 1951. Known for integrating Japanese traditional culture into its performances, Warabi-za focuses on regional folklore, history, and customs, especially from the Tohoku region. Their productions highlight and preserve local culture while engaging modern audiences through musical theater. This research program was developed and supervised by Warabi-za.)
Point 7: Methods
Clearly state the inclusion and exclusion criteria in bullet points for easier reading.
Response 7: Thank you for supportive reviews comments. According to reviewers suggestion, we have modified inclusion and exclusion criteria.
(Participants were required to be able to walk independently and live at home without personal assistance. Exclusion criteria included "dementia", "major depression", “severe hearing or visual impairment", "stroke", "Parkinson’s disease", "other neurological disorders", "intellectual disability", "certification for long-term care support due to disability under the Japanese public long-term care insurance system", and "inability to complete cognitive tests at the baseline assessment". )
Point 8: Provide more details on how the sample size calculation was conducted using G*Power.
Response 8: Thank you for supportive reviews comments. According to reviewers suggestion, we have modified more details on how the sample size calculation.
("Using G*Power to perform a sample size calculation for independent-samples t-tests (comparing means between control and intervention groups), with an allocation ratio of 4:1 favoring the control group, it was determined that a sample size of 160 participants in the control group and 40 participants in the intervention group would be required to detect a clinically meaningful difference between the groups. This calculation is based on a two-tailed test, with a significance level (α) of 0.05, a statistical power of 80%, and a medium effect size (Cohen's d) of 0.50.")
Point 9: Include a brief rationale for why the theater intervention was chosen and how it was expected to impact the participants.
Response 9: Thank you for supportive reviews comments. According to reviewers suggestion, we have added information
of theatre intervention.
(Theatrical interventions led by theatre professional have been shown in numerous studies to effectively improve both the physical and mental functions of elderly indi-viduals. Engaging in theatrical activities involves movements and vocal expressions that require full-body use, making it a form of exercise that enhances physical abilities. Specifically, improvements in balance, muscle strength, and flexibility can be expected, which help reduce the risk of falls and improve daily functional mobility. Moreover, theatrical activities stimulate cognitive functions as they involve memorizing lines, choreographed movements, and emotional expressions. This process helps enhance memory and concentration, preventing cognitive decline. The need for spontaneous dialogue and decision-making during rehearsals and performances also sharpens attention and flexibility in thinking.On the psychological side, theatrical interventions offer participants opportunities for emotional expression and social interaction, contributing to emotional relief and stress reduction. Taking on roles in performances boosts self-esteem and self-worth, enhancing life satisfaction and giving participants a greater sense of purpose.)
Point 10: Adjust Scheme 1 and reorganize Figure 1 to present them as a single, cohesive figure.
Response 10: Thank you for supportive reviews comments. According to reviewers suggestion, we have reorganized Figure 1.
Point 11: Results
Add a brief description of the participants' demographics and any baseline comparisons to highlight the homogeneity between groups.
Response 11: Thank you for supportive reviews comments. According to reviewers suggestion, we have added description of the participants' demographics and any baseline comparisons.
(Participant characteristics were compared to emphasize homogeneity between groups at baseline. The results showed a trend toward lower age and higher independence in the intervention group, but none of the differences were significant (Table 1))
Point 12: Ensure all p-values are reported with exact values, and include effect sizes where applicable to provide more insight into the magnitude of the findings.
Response 12: Thank you for supportive reviews comments. According to reviewers suggestion, we have added effect sizes.
Point 13: Include more details in the table captions and figure legends to enhance understanding without having to refer back to the text.
Response 13: Thank you for supportive reviews comments. According to reviewers suggestion, we have added include more details in the table captions and figure.
Point 14: Discussion and Conclusion
Provide more detailed comparisons with similar studies to strengthen the context of our results.
Response 14: Thank you for supportive reviews comments. According to reviewers suggestion, we have included comparisons with similar studies in our discussion.
Point 15: Offer a more detailed explanation of the limitations and how they might be addressed in future research. Suggest more specific areas for future research, such as the impact of different types of theater-based interventions or long-term follow-up studies.
Response 15: Thank you for supportive reviews comments. According to reviewers suggestion, we have described in detail the limitations of this study and added information on future developments.

Reviewer 2 Report
Comments and Suggestions for Authors
This manuscript aims to clarify the effects of a theatre training program intervention on the physical and cognitive functions in community-dwelling older people. It is helpful to improve the physical and cognitive functions for the elderly. To strengthen the article, the authors should consider the following:
1. Abstract: “Of the 75 participants, 30 were the control group, and 29 were the intervention group(page 1, line 18)”, the number of participants described here is missing 16.
2. Abstract: “and a significant interaction was found for GDS-15 in ANOVA (F=5.76, p<0.05) (page 1,line 24)”,the F value is different from the value in Table 2. Please confirm which one is correct.
3. Introduction: the authors mainly describe the definition and interventions of frailty. The theoretical basis of a senior theater project intervention in frailty should be provided. At the same time, it is suggested to provide a review of the research on the outcome of theater intervention.
4. Materials and Methods:“Based on G*Power for t-tests (means, difference between the control and intervention groups) with an allocation ratio of 4:1 for the control group/intervention group, it was estimated that a sample of 160 persons in the control group and 40 persons in the intervention group…(page 3,line 101-104)”Frailty is qualitative data. Is it appropriate to calculate the sample size only with mean, standard deviation, etc.? Is it appropriate to include only 30 controls in this study?
5. Materials and Methods: What is the applicability of NCGG-FAT to the elderly? Reliability and validity results should be reported.
6. Results:“Next, the UWS and GDS-15 had an interactive effect between the group and the time factors (Group Time interactions) (Figure 5) (page 6,line 196-198).” However,figure 5 shows only the interactive effect of UWS.
7. Discussion:the public health value of this manuscript should be added in the discussion.
Author Response
Response to Reviewer 2Comments
Point 1: Abstract: “Of the 75 participants, 30 were the control group, and 29 were the intervention group(page 1, line 18)”, the number of participants described here is missing 16.
Response 1: Thank you for supportive comments. As the reviewer suggested, we have revised the number of participants number.
(Of the 59 participants, 30 were the control group, and 29 were the intervention group.
)
Point 2: Abstract: “and a significant interaction was found for GDS-15 in ANOVA (F=5.76, p<0.05) (page 1,line 24)”,the F value is different from the value in Table 2. Please confirm which one is correct.
Response 2: Thank you for supportive comments. As the reviewer suggested, we have added revised F value in Table 2 (Page3)
Point 3: Introduction: the authors mainly describe the definition and interventions of frailty. The theoretical basis of a senior theater project intervention in frailty should be provided. At the same time, it is suggested to provide a review of the research on the outcome of theater intervention.
Response 3: Thank you for supportive comments. As the reviewer suggested, we have added a rationale for theatre intervention against frailty.
Point 4: Materials and Methods:“Based on G*Power for t-tests (means, difference between the control and intervention groups) with an allocation ratio of 4:1 for the control group/intervention group, it was estimated that a sample of 160 persons in the control group and 40 persons in the intervention group…(page 3,line 101-104)”Frailty is qualitative data. Is it appropriate to calculate the sample size only with mean, standard deviation, etc.? Is it appropriate to include only 30 controls in this study?
Response 4: Thank you for supportive comments. As the reviewer suggested, the number of subjects in this study was too small. Therefore, we have added the information to the limitations of this study of consideration.
(Second, the sample size of this study was relatively small, with 30 participants in the control group and 29 in the intervention group, which limited the statistical power of the study. For example, although there were trends in the improvement of cognitive function, these differences were not statistically significant. A larger sample size may have allowed for more robust conclusions, particularly in detecting smaller or more nuanced effects.)
Point 5: Materials and Methods: What is the applicability of NCGG-FAT to the elderly? Reliability and validity results should be reported.
Response 5: Thank you for supportive reviews comments . As suggested by the reviewer, we have added details of NCGG-FAT.
(NCGG-FAT is a tool developed to assess cognitive functional health in older adults. It has been validated for reliability and validity in various studies. It correlates well with other established functional assessment tools and accurately reflects older adult's cog-nitive functional status, supporting its use in both clinical and research settings [25].)
Point 6: Results:“Next, the UWS and GDS-15 had an interactive effect between the group and the time factors (Group Time interactions) (Figure 5) (page 6,line 196-198).” However,figure 5 shows only the interactive effect of UWS.
Response 6: Thank you for supportive reviews comments. As suggested by the reviewer, we have added figure of GDS-15.
Point 7: Discussion:the public health value of this manuscript should be added in the discussion.
Response 7: Thank you for supportive reviews comments. According to reviewers suggestion, we have added public health value of this manuscript.
(The manuscript highlights several important public health benefits of the theatre intervention for older adults, especially in preventing frailty and improving both physical and mental health outcomes. Theatre activities foster social interaction and emotional expression, which can mitigate the effects of social isolation, a common issue among the elderly. Social isolation is linked to various negative health outcomes, including increased mortality risk, and the study suggests that the intervention strengthens social bonds, contributing to both emotional well-being and overall life satisfaction. The manuscript provides strong evidence that integrating artistic and social activities into community-based health programs for the elderly could play a significant role in public health strategies aimed at enhancing both physical and mental well-being.)

Round 2
Reviewer 1 Report
Comments and Suggestions for Authors
This manuscript is improved and accepted in current form.
Reviewer 2 Report
Comments and Suggestions for Authors
I don't have any other comments.